# Sustainable Utilization of Fishery Waste in Bangladesh—A Qualitative Study for a Circular Bioeconomy Initiative

**Mohammad Mojibul Hoque Mozumder [1,*]**, **Mohammad Muslem Uddin [2]**, **Petra Schneider [3]**, **MD Hadiul Islam Raiyan [2]**, **Most. Gulnaher Akhter Trisha [2]**, **Tabassum Hossain Tahsin [2]** and **Subeda Newase [4]**

[1] Helsinki Institute of Sustainability Science (HELSUS), Fisheries and Environmental Management Group, Faculty of Biological and Environmental Sciences, University of Helsinki, 00014 Helsinki, Finland
[2] Department of Oceanography, Faculty of Marine Sciences and Fisheries, University of Chittagong, Chittagong 4331, Bangladesh; mmu_ims76@cu.ac.bd (M.M.U.); hadiul.imsf@gmail.com (M.H.I.R.); gulnaher.trisha@gmail.com (M.G.A.T.); taseentabassum656@gmail.com (T.H.T.)
[3] Department for Water, Environment, Civil Engineering and Safety, University of Applied Sciences Magdeburg-Stendal, Breitscheidstraße 2, D-39114 Magdeburg, Germany; petra.schneider@h2.de
[4] Faculty of Fisheries, Chattogram Veterinary, and Animal Sciences University, Chattogram 4225, Bangladesh; shishirsubeda6184@gmail.com
\* Correspondence: mohammad.mozumder@helsinki.fi

**Abstract:** Living marine resources are limited; therefore, utilizing them sustainably is essential. Not all resources obtained from the sea are used adequately, causing discards, on-board waste, and by-products and waste on land. Recognition of the limited marine resources and increasing environmental pollution has emphasized the need for better utilization of by-products. Waste may include particles of flesh, skin, bones, entrails, shells or liquid stick water. Unfortunately, no fishery waste and by-product management initiatives or projects exist in Bangladesh; by-products are generally thrown into dustbins, ponds, rivers, and the sea. Bangladesh's fish-processing waste and fishery by-products could be exported, providing a source of foreign currency earnings. Primary and secondary data were collected through documentary analysis, a literature review, and in-depth interviews (*n* = 129) with fishers and other relevant stakeholders regarding the challenges of Bangladesh's sustainable fishery by-products and fish-processing waste. The data were analyzed thematically, guided by the most meaningful stories, and show that fish waste, or fishery by-products, should not be considered less valuable than the fish itself but is a precious and profitable resource capable of bringing health, social, economic, and environmental benefits. Our results reveal that fishery waste can expand local communities', especially fishers' and other workers', potential for jobs or alternative income-generating tasks during fishing ban seasons. Finally, suggestions for managing fishery waste and fishery by-products are made to ensure improved and sustainable utilization via a circular bioeconomy.

**Keywords:** fishery by-products; seafood processing waste; sustainability; profitability; bioeconomy initiative



## 1. Introduction

Fish is an essential source of nutrients, including animal protein, for human health worldwide [1]. However, not all fish catches from waterbodies are used adequately, and three differentiated types of waste are produced: discards, waste on board, and by-products and waste on land [2,3]. In recent years, the world's fishery resources have exceeded 160 million tonnes. However, a considerable amount of the total catch is discarded every year as by-catches or processing leftovers, including trimmings, fins, frames, heads, skin, viscera, and others [4]. In addition, many processing by-products, such as crustacean and shellfish shells, are accumulated from marine bioprocessing plants. Sustainable utilization of by-products is vital considering the limited marine resources and the increasing environmental pollution worldwide [5].

Marine by-products contain valuable protein and lipid fractions, minerals, and enzymes. Despite the low profitability, a significant fraction of by-products is used for fish feed production, including making fishmeal and fish oil [6]. Nevertheless, there are many ways in which fish and shellfish waste could be better utilized, including the production of novel food ingredients, nutraceuticals, pharmaceuticals, biomedical materials, fine chemicals, and other value-added products [7].

The definition of "fish waste" includes many fish species or by-catch products with no or low commercial value and undersized or damaged commercial species [8]. Fishery waste also comprises the offal generated when processing fishes at sea and discarded undersized fishes and non-target species [9]. The amount of fishery waste produced is extensive: approximately 7.3 million tonnes of discards are returned to the sea annually by fisheries worldwide [10]. It has been projected that more than 50% of fish tissues, including fins, heads, skin, and viscera, are discarded as they are considered to be waste [8]. The annual discards from the world's fisheries exceed 20 million tonnes, and they include "non-target" species, fish-processing waste, and by-products. However, the overall amount of discards is highly variable as the amounts and composition of species of fishery waste vary between different fishing areas [11,12].

Fish preservation and processing vary according to the species because they have a characteristic composition, size, shape, and intrinsic chemistry [13]. However, according to the FAO, around 7.3 million tonnes of whole fish (~8% of the global catch by volume) are discarded before landing worldwide every year in commercial fisheries [14]. Generally, the discards from edible operations are conservatively estimated at 50% of whole fish but can range from 10% to 90%, depending on the fish species and the intended use [15]. A vast amount of biomass is discarded and is generally incinerated, increasing the energy consumption, financial cost, and environmental impact of their management process, or is utilized for low-value products; to date, fish waste is used mainly in the fishmeal industry since it contains almost the same amount of proteins as fish meat [16]. Moreover, the nutritional composition of fish waste enables it to supply plant nutrients or enrich compost. Fish waste can be processed to produce numerous fertilizers, and commercial fish-based fertilizers are currently used for agricultural and horticultural crops [17]. In addition, fish waste has a high concentration of biodegradable organics, which could be recycled as an attractive co-substrate for waste-activated sludge to improve methane production during anaerobic co-digestion [18].

Fish-processing operations generate potentially substantial quantities of waste and by-products from inedible fish parts and endoskeleton shell parts from the crustacean-peeling process, such as particles of flesh, skin, scales, bones, visceral mass (viscera, air bladder, gonads, and other organs), head, fins, shells or liquid stick water [19]. The volume and concentration of wastewater from fish processing depend on the raw fish composition, the additive used, the processing water source, and the unit process. The management of this fishery waste is one of the main problems, arousing the most significant concern and having the strongest impact on the environment [20]. The runoffs, typically high in nutrients, result in algal blooms, unpleasant odors, acutely lethal discharges, and localized areas of anoxia [21].

A by-catch is the non-intended capture of non-target fish species. Some species are retained for sale, while others are thrown back into the sea due to their low value or the legal requirements [3]. The ecological impacts of by-catches are significant as they can change the availability of prey, affecting marine ecosystems and the productivity of fisheries [9]. In addition, there has been a remarkable increase in the amount of fish waste produced worldwide; it has been estimated that about two-thirds of the total amount of fish is discarded as waste, creating enormous economic and environmental concerns [22]. Hence, the disposal and recycling of this waste have become a pivotal issue to be resolved, and the concepts of the circular bioeconomy can be helpful in this regard [23].

Our present economic system fails to appreciate nature due to its linear approach, which fosters waste generation and disposal. Hence, innovative economic models based on

a cradle-to-cradle approach are needed [24]. The bioeconomy encompasses the production of renewable biological resources and the conversion of these resources and waste streams into value-added products, such as food, feed, bio-based products, and bioenergy [25]. Conversely, the circular economy is presented as the economic space where the value of products, materials, and resources is maintained for as long as possible, minimizing waste generation [26]. The concepts of the bioeconomy and circular economy have similar targets. They can therefore be merged in the ideal case into a circular bioeconomy approach that addresses the circular management of biobased material flows [27].

In this case, the circular bioeconomy offers a sustainable development plan with a creative framework for using natural resources to achieve a sustainable economy [28]. The benefits of the circular bioeconomy include improved resource and eco-efficiency, lower greenhouse gas emissions, reduced reliance on fossil resources, and valorization of side and waste materials from numerous sources, such as agro-industrial aquaculture and fishery [29].

The circular bioeconomy aims to bring together environmental conservation and poverty alleviation to improve human well-being and social equity while significantly reducing environmental risks and ecological scarcities [30]. The greatest strengths of the circular bioeconomy are awareness among people and industry, the involvement of stakeholders and policymakers, the support of politics, sustainable production and consumption, resource valorization, and zero waste [31]. In this sense, the bio-waste valorization approach plays a fundamental role in bringing circularity to the bioeconomy. However, the success of the circular bioeconomy requires modern technology, innovation, and knowledge of tradition [32]. Finally, with the growing attention to the circular economy, the exploitation of underused or discarded marine material can represent a sustainable strategy for realizing the circular bioeconomy, with the production of materials with high added value [33]. There is a lack of initiatives to utilize the vast amount of fish waste in Bangladesh. A circular bioeconomy approach in this respect may play a vital role in sustainable fish-waste management.

Bangladesh has become a self-sufficient fish-producing country and provides about 60% (with of 62.58 g/day per capita against the targeted 60 g/day) of the population's total daily animal protein intake [34]. Bangladesh earns huge amounts of foreign currency by exporting fish, shrimp, and other fishery products. The fisheries sector contributes 1.39% to the national export earnings [35]. Fish is the second most valuable product in Bangladesh, and its production contributes to the livelihoods and employment of millions of people. Therefore, the culture and consumption of fish have important implications for Bangladesh's national income and food security [36].

The fisheries sector in Bangladesh can broadly be divided into four sub-sectors—inland capture, inland culture, mariculture (artisanal fisheries), and marine industrial fisheries. The country has a coastal area of 2.30 million hectares and a 714 km coastline along the Bay of Bengal and supports large numbers of artisanal and coastal fisheries [37]. Bangladesh presently ranks fifth in the world for aquaculture production and produced over 4.503 million (45.03 lakh) tonnes of fish during the fiscal year (FY) 2019–2020. Aquaculture accounts for 57.38% of the total fish production. It is expected that the country will exceed its production target of over 4.552 million (45.52 lakh) tonnes of fish by FY 2020–2021 [34].

Bangladesh's total output of 4.4 million tonnes of fish in 2020 was sourced from inland capture (28.45%), inland culture or farmed fish (56.24%), and marine capture (15.31%) [38]. The coastal and marine fisheries of Bangladesh are very diverse. Bangladesh's marine fisheries resources, however, remained largely untapped. There is a total of 166,000 km$^2$ of water area, including the Exclusive Economic Zone (EEZ) in the Bay of Bengal in the south of Bangladesh, but fishing is confined to a depth of 200 m. As a result, most domestic fish consumption is of freshwater species. This was originally from inland capture fisheries. Inland fisheries comprise rivers, ponds, estuaries, floodplains, and brackish water [39].

Two types of aquaculture practices take place in Bangladesh: freshwater and coastal aquaculture. No marine aquaculture is currently practiced in the country, and no marine

or coastal finfishes are farmed. Freshwater aquaculture comprises pond farming of carps (indigenous and exotic), Mekong pangasid catfish, tilapia, Mekong climbing perch, and several other domesticated fish, though on a smaller scale. Coastal aquaculture consists mainly of shrimp and prawn farming in ghers (coastal pond or enclosures) [40]. Bangladesh exports 10 main categories of fishery products (frozen freshwater fish, frozen marine water fish, frozen shrimp, chilled fish, live fish, dry fish, salted dehydrate, live kusia, live crabs, and fish scales/shrimp skulls) to more than 55 countries [40].

Fish landing centers accumulate different types of fresh fish and fisheries commodities from different harvesting sources, such as rivers, ponds, estuaries, and the sea. The harvested fish are transferred from the landing centers to the consumer markets via different channels [41]. Most of the landing centers are in the areas where a substantial quantity of fish is produced. Conversely, most large fish markets are in district cities and municipal areas where large numbers of affluent people are concentrated. The major available fish in the landing centers belong to five main orders, namely Clupeiformes, Cypriniformes, Siluriformes, Perciformes, and Channiformes. So far, 40 species have been identified as significant available species, and they belong to 32 genera and 15 families. A total of 237 fish landing centers and 5440 markets exist in Bangladesh [42,43].

Despite the importance of small-scale fisheries, the management of coastal fisheries in Bangladesh has focused predominantly on industrial trawler fleets, with limited attention being paid to others in the sector. This has led to an uncontrolled expansion of fishing efforts, exacerbating the crisis. Artisanal fishing has already become non-remunerative. Poor fishers are setting more and more fine-mesh nets to survive, resulting in excessive pressure on fish stocks [44]. Bangladesh has 130 deep-sea fishing trawlers, 22,000 mechanized fishing boats, and 25,000 non-mechanized fishing boats [45]. In total, there are 133 fish-processing plants in Bangladesh, primarily located in port cities (Khulna and Chittagong), 74 of which are EU approved [46]. Furthermore, according to the Bangladesh Shrimp and Fish Foundation (BSFF), 71% of the fish is consumed and 29% is wasted [47]. At the moment, there are no fish-waste management initiatives or projects in Bangladesh. Generally, fish waste and by-products are thrown into dustbins, pond, rivers, and seas.

In Bangladesh, by-catch and fishery by-products and fish-processing waste could be new items for exporting and a source of foreign currency earnings. The proper management and utilization of fishery waste and its conversion into value-added products will lead to better resource utilization and profit maximization and result in significant environmental and economic improvement [48,49]. In contrast, underutilization of by-products leads to potential revenue losses and additional disposal costs [17]. Some studies have focused on seafood and livestock waste management in Bangladesh [50–53].

Every year, a vast amount of fish waste is generated in Bangladesh. According to the Fisheries Department, Bangladesh generates 93 thousand tonnes of fish-processing waste a year. Only a small portion, 900 tonnes, is used [53]. Due to the nutritional properties of fish waste and its volume, there is considerable potential for circular bioeconomy approaches as a base for sustainable fish waste management that might be considered as a closed protein cycle in the long term [54]. However, there is scant information about the sustainable utilization of fishery waste and fishery by-catches, including the marketing, processing, and uses of fish, in Bangladesh. Based on a synthesis of secondary and primary data, assessed using qualitative methods (in-depth individual interviews), the present study poses the following research questions:

- What is the present status of fishery waste and fishery by-catches and of their environmental impacts in Bangladesh?
- What are the potential uses and challenges of fishery waste for sustainability?

## 2. Materials and Methods

Most fish-processing industries are located in three coastal districts of Bangladesh, namely Khulna, Chittagong, and Cox's Bazar. Therefore, Khulna (Batiaghata, Paikgacha, and Rupsa), Chittagong (Baizid, Halishahar, Khulshi, Kotwali, Pahartali, Panchlaish, and

Patenga), and Cox's Bazar (Sadar Upazila and Kutubdia) were selected as study areas (Figure 1). These chosen study areas offer high potential for fresh and brackish water capture and culture fisheries, including vast marine resources. The interviewees were from different locations in the districts (fish landing centers, fishers' households, fish markets, fish-processing industries, dry fish yards, and government offices).

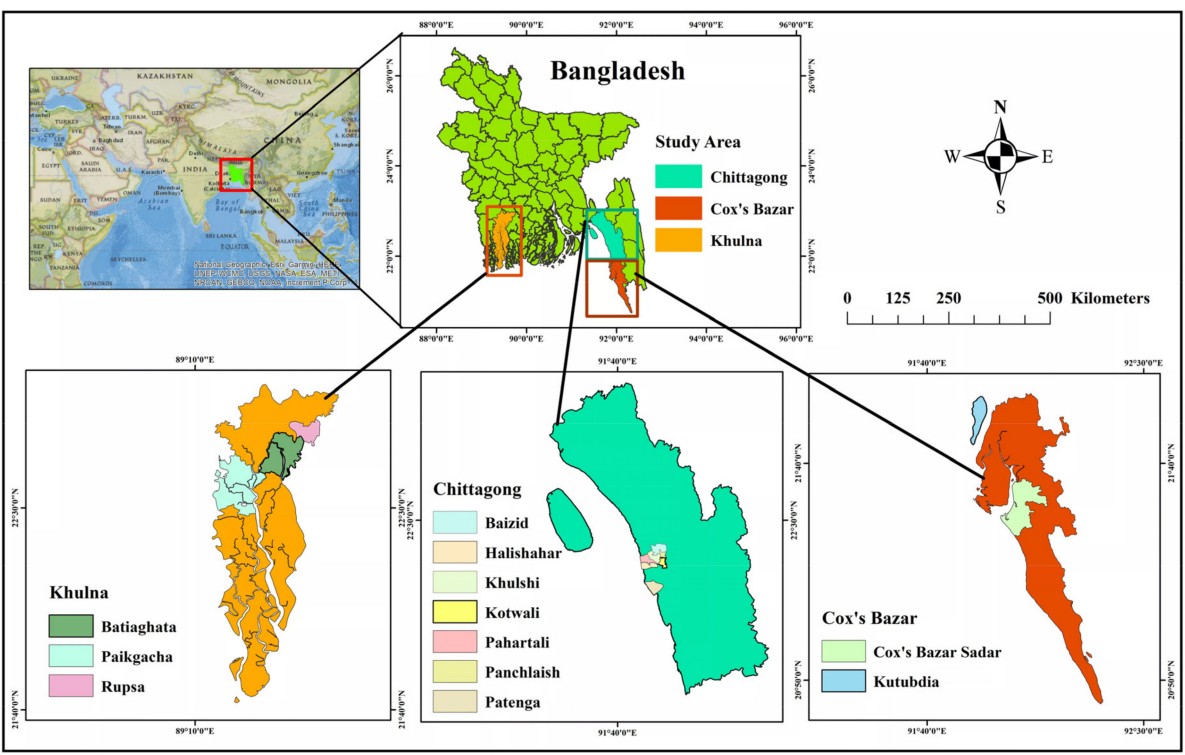

**Figure 1.** Location of the study areas.

The present study is based on a synthesis of secondary and primary data, assessed using qualitative methods. We used the qualitative method to answer questions about experience, meaning, and perspective—most often from the participants' standpoint [55]. Beyond that, we chose to follow a qualitative methodology here because it enabled us to consider our interviewees' various areas of expertise more flexibly. Since sustainable fishery waste utilization has yet to become fully operationalized in Bangladesh, we cannot yet analyze its statistical ramifications quantitatively, and thus we adopted a qualitative methodology somewhat by default.

Primary and secondary data were collected for the study from January 2021 to March 2021. In-depth interviews, as a qualitative method, were conducted to collect the primary data. In-depth interviews are useful when detailed information about the respondents' thoughts and behaviors is sought or the aim is to explore current issues in depth [56]. Using a semi-structured questionnaire (see the Supplementary Materials), a total of 129 in-depth individual interviews (Table 1) were conducted. The respondents were available fishers ($n = 30$), fishing crew members on fishing vessels ($n = 15$), processing crew members on board ($n = 6$), custodians at fish landing centers ($n = 3$), fish sellers ($n = 15$) and fish cutters ($n = 12$) in fish markets, managers ($n = 6$) of fish waste-processing centers, dealers ($n = 9$), suppliers ($n = 6$), and exporters ($n = 3$) engaged in the fish waste business, and quality control officers ($n = 6$) in seafood-processing centers. Together with those stakeholders, we interviewed academics ($n = 9$), district fishery officers ($n = 3$), district environment officers ($n = 3$), and marine fisheries department officials ($n = 3$).

**Table 1.** Sample of interviewed participants.

| Participants/Stakeholder Groups | Number of Participants in Chittagong | Number of Participants in Cox's Bazar | Number of Participants in Khulna |
|---|---|---|---|
| **Fishers** | 10 | 10 | 10 |
| **Fishing crew on fishing vessels** | 5 | 5 | 5 |
| **Processing crew on board** | 2 | 2 | 2 |
| **Fish landing centers** (responsible officers) | 1 | 1 | 1 |
| **Fish markets** | | | |
| Fish sellers | 5 | 5 | 5 |
| Fish cutters | 4 | 4 | 4 |
| **Fish waste-processing center** (manager) | 2 | 2 | 2 |
| **Fish waste business** | | | |
| Dealer | 3 | 3 | 3 |
| Supplier | 2 | 2 | 2 |
| Exporter | 1 | 1 | 1 |
| **Seafood-processing industry** (quality control officer) | 2 | 2 | 2 |
| **District environmental office** (environment officer) | 1 | 1 | 1 |
| **Marine Fisheries Department officials** | 1 | 1 | 1 |
| **Fisheries Department** (district fishery officer) | 1 | 1 | 1 |
| Academics | 3 | 3 | 3 |

We used the purposive and snowballing sampling strategies to select the interview respondents. The purposive sampling approach was employed to interview more knowledgeable fishers and other stakeholders [57], and the snowballing sampling method was used to identify potential fishers to interview because of the diverse group of people engaged in fishery waste generation [58]. In-depth individual interviews provided us with an understanding of each interviewee's personal and cultural perspectives, life experiences, and economic circumstances as expressed in their own words [59]. The interviews were semi-structured, allowing for free-flowing conversation as well. At the beginning of the interviews, the interviewees were presented with the idea of fishery waste, fishery by-catches, fish-processing waste, and their utilization as a conversation starter. They were then asked what they knew or thought about the status of the fishery waste and fish-processing waste and their utilization in Bangladesh and whether they felt that allocating resources to the promotion of such valuable things in Bangladesh might be worthwhile as a circular bioeconomy. They were then asked for their perspectives on the potential benefits and challenges of implementing fishery waste, fish processing, and by-product utilization in Bangladesh. The conversation was directed toward the types of initiatives that the government and other agencies should implement to overcome such challenges and the management strategies that might be implemented for fishery by-catches and fish-processing waste and their utilization in Bangladesh. The interviews were carried out in the local dialects and language (Bengali). Each interview lasted an average of one hour.

We shared the intention behind the data collection and obtained verbal informed consent. Before each interview, we informed the participants about the study and assured them about its ethical principles, including the rights to anonymity and confidentiality. Prior permission was obtained for all recordings of interviews and photographic documentation. Participation in the research was entirely voluntary, and all the participants were informed of their right to withdraw from the study at any stage. To obtain secondary information, a literature review related to fishery by-catches, fishery waste, and seafood-processing waste and their sustainable utilization as well as the relevant policy documents in Bangladesh and other countries was conducted using online search engines (see the Supplementary Materials).

The thematic analysis method was used for the data analyses. We employed six steps of the inductive approach to thematic analysis [60] as follows: familiarizing oneself with the data, generating initial codes, searching for themes, reviewing the themes, defining and naming the themes, and producing the report/manuscript. Next, we clustered the interview material according to the issues that stakeholders felt should be considered in utilizing fishery by-catches and fishery waste. As a result of the thematic analysis, the following data categories were formed: (i) the present status of fishery by-catches and fishery waste utilization in Bangladesh; (ii) the potential for fishery by-catches and fishery waste, including fish-processing waste, in Bangladesh; (iii) the possible environmental impacts of fishery waste; (iv) the practical challenges entailed in the sustainable utilization of fishery waste; and (v) a management strategy to enhance the sustainable utilization of fishery waste as a circular bioeconomy in Bangladesh. Finally, we presented the key findings from our empirical material, using direct quotations that exemplify the categories into which portions of the interview transcripts were clustered. No qualitative data analysis software was used in this study; the entire analysis was carried out manually.

## 3. Results

### 3.1. Socio-Economic Profile of the Respondents

The average age of the individual fishers, fish sellers, fish cutters, and waste-processing center staff was 40, the oldest respondent being 85 years old (Table 2). The average fishing experience of the fishers was 15 years, with the greatest experience being 41 years. The average monthly income ranged from BDT 17,500 (USD 1 USD = BDT 86) to around BDT 31,000. The daily relations with waste were expressed as binary values, 1 meaning that the respondents were directly involved with fisheries' waste and 0 meaning that they were indirectly involved with fisheries' waste. Most of the fish cutters and waste-processing center staff work for an average of 8 h a day, though some spend up to 20 h daily on waste processing.

**Table 2.** Socio-economic profile of the respondents.

| Variables | Definitions | Mean (Min., Max.) |
|---|---|---|
| Age (years) | Age of individual fishers, fish sellers, fish cutters, waste-processing center staff and owners | 40 (18, 85) |
| Educational qualifications | Number of years of formal schooling | 5 (1, 10) |
| Years of experience | Number of years of waste processing/cutting/selling experience | 15 (41, 2) |
| Average monthly income | Monthly income from fisheries waste | 17,500 (12,000, 31,000) BDT |
| Fisheries waste daily relations | Binary indicator with 1 meaning involved with fishery waste and 0 meaning indirectly involved with fishery waste | 1 (0, 1) |
| Working hours | Fishers' total number of working hours in a day | 10 (8, 11) |

### 3.2. Present Status of Fishery Waste and Fishery By-Products in Bangladesh

Several studies have been conducted on the utilization of fish scales from seafood-processing waste in Bangladesh (see the Supplementary Materials). Although data are available about the fish production per year in Bangladesh, there is a lack of data on the fish waste generated per year in Bangladesh. There is a rough estimation of 93 thousand metric tons of yearly fish-processing waste products in Bangladesh. However, a small portion, 900 tonnes, is used to produce different things [61]. The seafood industry in Bangladesh produces around 43,320.88 tonnes of seafood waste per year, and its value of USD 13.73 per tonnes totals USD 44.09 million. The largest amounts of both fish and shrimp waste are produced in Khulna, followed by Chittagong, Cox's Bazar, Dhaka, and Sylhet. Annually, the fish markets in Khulna and Chittagong produce over 2500 tons of

waste, which is generated from retailers' purchases [53]. Based on primary and secondary data, we summarized the general uses of fishery waste and fishery by-products and their present uses in Bangladesh, followed by recommendations for the sustainable utilization of such resources (Table 3).

**Table 3.** Summary of fishery waste and fishery by-products' general uses and their present uses in Bangladesh.

| Fishery Waste Product | General Uses | Present Situation/Uses in Bangladesh | Recommendations | Images (The Images Were Taken during the Conducting of Fieldwork via In-Depth Interviews) |
|---|---|---|---|---|
| Air bladder | To make surgical yarn and capsule cover caps [53]. | Larger sizes are imported to China, Thailand, Malaysia, and smaller ones are used in poultry feed. | Develop the pharmaceutical industry to utilize them properly. |  |
| Viscera | Used as feed for various types of fish, such as catfish, and the poultry industry [48]. | Widely used in the poultry and fish feed industry. | Feedstuff in fish and animal dietsas a protein substitute, for example fish silage and fishmeal. |  |
| Skin | To replace burnt skin in the human body [62]. | Exported to China, Vietnam, and Thailand. | Empirical research is needed so that it can be used widely in the medical arena in Bangladesh. | Not found |
| Fish oil | Used as a raw material for cattle, poultry, and fish feed in feed manufacturing industries [63]. | Used in fish feed and animal feed and exported. | Purified as edible oil and used in biodiesel. |  |

**Table 3.** *Cont.*

| Fishery Waste Product | General Uses | Present Situation/Uses in Bangladesh | Recommendations | Images (The Images Were Taken during the Conducting of Fieldwork via In-Depth Interviews) |
|---|---|---|---|---|
| Fish silage/powder | Used mainly in fish feeds and moist feed pellets [64]. | Used in animal feed and fish feed. | Can be used in animal feed as hydrolysate and in poultry feed. |  |
| Fish scales/bones | Waste fish bones modified with potassium hydroxide (KOH) are utilized as a cost-effective catalyst in the transesterification of refined, bleached, and deodorized (RBD) palm oil with methanol [65]. | Exported, fishmeal, and landfill. | Fashion designers can use them as ornaments. |  |
| Fish collagen | Used in the cosmetics industry for protecting skin moisture, increasing bone density, and strengthening immunity [66]. | Fish feed and pharmaceuticals. | Empirical research is needed for various uses. |  |

The qualitative interviews indicated both prospects for and constraints on the sustainable utilization of fishery by-catch products and fishery waste in Bangladesh. However, we have organized the results according to the objectives of the study in the following.

During the in-depth interviews, a senior scientist from the Department of Fisheries stated that there was no scope to collect such data due to various limitations. However, the scientist expressed the importance of such data collection for the sustainable utilization of fishery waste, including further research to speed up the circular bioeconomy process in Bangladesh. He concluded as follows: "This huge amount of waste is our burden. We can use only 900 tons of it. We fail to take advantage of the rest of the waste. The price of one tonne of waste in the international market is one thousand US dollars. If we take appropriate steps, we can earn a huge amount of foreign currency by selling this waste. However, for that, the government has to take steps including the right policy support."

A fish cutter, aged 43, from a local fish market in the study area of the Khulna region said: "I am illiterate and do not have knowledge about fishery waste and fishery by-products. My livelihood depends on fish cutting in this local market, and I have been processing the fish for the last 14 years as per customer instructions. I just take out the scales, fins, and other parts of the fish that are not edible. Nevertheless, I can say that,

although they are waste parts of the fish, they have some economic or market value. I heard that fish waste is used to produce different things. You see, I store the fish scales, air bladder, and intestines of the fish. I will sell them to the local agents later, and the price is approximately 100 BDT/kg. So, you see, it is an extra income for me" (USD 1 = BDT 85).

One academic from the University of Chittagong stated, during his interview about the present status of fishery waste and fishery by-products, that "Once upon a time fisheries waste was dumped; now an awareness that we know or do not know has already developed a concept that it can be reused. Its first use is in the case of fish feed or poultry feed. No value addition or quality maintenance is done here; it is used directly here. It is collected from the market or different points, dried in the sun in a large yard or an open slap. It is powdered and fed with food. There are different unhygienic processes that are unsuitable or recommended for making/manufacturing quality feeds."

A manager (aged 49) of a fish waste-processing center from Chittagong said, "Of course, its [fish waste] market value will be much better. Because if it could be preserved properly, that would be a good organic product. Moreover, everyone is interested in news of organic products or natural products. And look! People are turning to the previously used natural product. So, it is not just a fish feed or something like that. It can also be used as a fertilizer, which is a very high-quality fertilizer. So, it has a lot of multiple uses, and these are well established. However, I do not know why we are not trying this here."

Another academic, from the University of Khulna, said, "I do not know the exact market value at present, but maybe such an industry or market has not developed in Bangladesh yet. However, we should be positive because it has found a wide demand abroad. We would obtain a strong economic figure if it can be properly used. The entrails of the fish that are cut in our house are used as animal food. So, we can collect this waste from fish markets, especially in Dhaka's and Chittagong's major markets. It is possible to produce new products by recycling. We can make new products, like fish feed and poultry feed, with the discarded parts of the fish, such as scales, crackers, fins, and entrails. Also, gelatin, chitin, new ornaments, and fish powder can be made using fishery waste. However, initiatives to set up such industries are still poor in Bangladesh."

A scientist from the Bangladesh Fishery Research Institute, in Cox's Bazar, offered his perceptions about the present status of fishery waste as follows: "Fish waste management in Bangladesh is becoming familiar to fishers and other stakeholders in some specific regions, such as Najiratek, Fishery ghat, Chawfaldani, and Nuniachara in Cox's Bazar. Many of the professors at different universities in Bangladesh are now working on it. Some local dry fish businessmen sell most of them, like scale, gut, and fecal waste, to Dhaka and Chittagong for poultry feed. In the case of large marine species, they extract liver oil and larger air bladders and skeletons and export them to Thailand, China, and Indonesia."

An official from the Department of Environment, in Cox's Bazar, stated, "Fish waste is those discarded parts which are not edible. Now there is no fish waste-processing center in Cox's Bazar. The environment department did not take any steps against dumping fish waste into the sea due to a lack of laws. If the government initiates any law against this kind of activity degrading the water quality, we can take legal action. As fish waste is biodegradable, we do not think much more about this kind of pollution, but we have specific laws in the case of non-biodegradable pollutants. Nevertheless, that waste dumped in the water gets decomposed, decreases the oxygen levels, and increases microbial activity, resulting in life-threatening danger to aquatic organisms. As our waste management system is indigent, there is a necessity to upgrade the waste management system by doing empirical research."

A production officer from a fish-processing industry in Chittagong said, "Still, we are selling fish waste produced during fish processing. However, we will personally dry this waste and feed fish after some days. The raw materials of this waste can be used as fish or poultry feed and can be used as cosmetics, medicine, and a huge source of protein after following some processes. For example, now prawn shrimp waste is used as a great source of protein; its exoskeleton parts are used for various purposes. After processing the large

fishes in these processing centers, the bladders, liver, and valuable bones of the fishes are sold to local merchants. Moreover, in our factory, we also do not export fish waste parts; rather, we keep them to utilize on our own."

The fishers who work as a crew in the fishing boat stated their perceptions of and knowledge about fishery waste in the following manner (Abdullah, Porimol, and Rahim, aged 45, 24, and 28, respectively, from the Khulna region)—"Well, in general, we know what fishery waste is. For example, the parts of a fish that are not suitable to eat are called fishery waste. Although we do not know about the sustainable utilization of the fishery waste, we have seen that some local entrepreneur collects the fish guts, scales, liver, fins, and air bladders from the local fish market and even from the fish-processing center. Later they sell them to the merchants and export them to other countries. Also, we heard that it can be used for making different valuable goods."

A supplier of fishery waste in the Cox's Bazar area shared his story about how he processes the fishery waste and sells it in the market: "Buying the fish waste from local sellers, I dry them up with the help of local women here and then sell them in Dhaka and Chittagong to the brokers. My main task is to dry them and sell them because many people make fish feed and medicine. Two types of powder are produced. One of them has an excellent value; it is sold in the market for 70–90 BDT per kg and used to prepare the fish feed. One of them is not as good value and is sold in the market for 60–70 per kg, and it is used for preparing the poultry feed. We generally buy the air bladders of the big fishes and use them to make fish feed."

*3.3. Measuring the Environmental Impacts of Fishery Waste*

While discussing the environmental impacts of fishery waste, one general fisher from Chittagong, aged 39, stated his perception in the following manner: "You know we are not educated, and we have only general ideas, and we can tell from our experiences. See, I have been fishing in the Karnafully River since I was 15. I have seen fish waste dumped into the river regularly. I am not sure whether it harms the environment or not. But, for sure, I can say that the water quality of the Karnafully is changed, I can smell the bad odor of the water, I cannot see the fish species that I used to see. Based on that, I can say that all the unwanted waste is responsible for this situation in the Karnafully River."

Regarding the environmental impact of fishery waste and by-products, one senior scientist from the Bangladesh Fisheries Research Institute, Khulna, stated during the interview that, "Undoubtedly, it has a harmful effect on the environment. Because it is not a matter of one day, it is coming over the years. Moreover, as the days go by, there will be the bio-accumulation process; day by day, it will increase. As it increases, so does the pollution rate, and what do we see? That is where they fall. It smells bad. Besides, it is responsible for different microbial growth. Furthermore, some specific microbes may be spread to the environment. Again, it can be bad for human beings. Secondly, various toxic gases are forming, whether aquatic fish die due to viral-bacterial or microbes. There will be different toxic gases from different toxic products. They will create toxicity, remaining in the water as a micronutrient. There are phytoplankton, zooplankton, different kinds of fish, shrimps, and crabs. What will happen again is that some animals will die to minimize this pollution. It is harmful to everyone. So, there is no chance of underestimating its impact. Its long-run process will also be dangerous."

An academic from the University of Khulna said, "The ocean can be polluted by fishery waste when fishery waste is thrown out into the water body. Fishery waste is organic matter, which is biodegradable and decomposed. So, the ecosystem is being hampered due to nitrate and phosphate formation for decomposition. Moreover, the production of ammonia causes toxicity to the environment. The ultimate result is harmful to the environment and the ocean. It causes the death of fish and plankton."

A crew member of a fishing vessel said, "In Bangladesh, fishing boats dump their waste into canals, rivers, or seas. The fishing boats do not have any garbage storage. They throw the dirt/garbage directly into the sea. There is no place to keep the waste in their

ship. If there is space, they bring the fish in it. They want to earn profit for expenses because they catch so many fish that they cannot keep the fish waste separate or throw it into the river. If a small fish is caught at the beginning, if they catch a large and high market price fish, then the cheap or small fish that was caught at the beginning is thrown into the sea."

A senior official from the Department of Environment, Chittagong, stated, "Well, you know, waste management is the prime concern of the environmental department at present in Bangladesh that is produced by different industries, including the fish-processing industry. The fish-processing industry is characterized by a high volume of water used in the production stages and, consequently, substantial effluents. If not treated, all those effluents find their way into the water body and harm the aquatic environment. So, there is a necessity to set up a waste management plant by the fish-processing industries in Bangladesh to make the aquatic environment free from pollution."

*3.4. Potential and Challenges of the Sustainable Utilization of Fishery Waste*

This section summarizes the respondents' perception of and knowledge about the potential and challenges of utilizing fishery waste sustainably. We also gathered information regarding the economic value of fishery waste and its possible integration into the circular bioeconomy process. During the interviews, an academic from Khulna University shared his opinion about the potential and challenges of sustainable fishery waste utilization as follows:

"I have been researching fish feed for the last ten years. Hence, I would like to talk about fish feeds from fishery waste, nutrition from it, recycling and minimizing pollution, and how I can reuse it and, in a sense, the circular bioeconomy approach. Each waste part or by-product has its reuse, including fishery waste. However, the main obstacle is that our collection process is unhygienic. It is not maintained correctly and scientifically, so it is challenging to get a quality product from here, such as hard parts like bone, it is being washed. However, it is essential to maintain the quality of the biodegradable, soft parts. If I treat them as waste, waste will never give us value-added products. Also, if I do not keep the biodegradable part at a lower temperature, there is constant decomposition; I can never get anything valuable from a decomposed product. So, I must ensure quality maintenance from the first step to the end." Later, he added, "if you maintain waste collection quality and segregate waste through industrial, scientific processing, which can be used to separate the work or the type of compound from which it can be produced, then it is possible to develop it very well industrially. In this case, many things will be possible at the same time. Firstly, we are getting quality products. Secondly, it is possible to add value. Thirdly, pollution control is being done."

There are opportunities for innovation from the fishery waste. A manager of a fish waste-processing center in Chittagong shared his perception and knowledge of the potential uses of fishery waste in Bangladesh: "You know, many products have already been produced from fish waste, it has been published in literature or articles, there are many available in the market, but we can't do extraction work like that in Bangladesh, from fish scale to cosmetics, from air bladder to pharmaceuticals (surgical yarn). Shark soup can be made from fin, which is one of the favorite dishes in China. Also, different types of ornaments and jewelry items can be made from bone or tooth; leather items can be made from fish scales and skin; chitin and chaetocin can be made from shells. So, it is necessary to do empirical research on producing different valuable things from the fishery waste and export them to other countries. It will enhance the employment opportunities for more people in Bangladesh and earn more foreign currency."

A senior scientist from the Marine Fisheries Department, Khulna stated his perceptions about the potential sustainable use of fishery waste in the poultry sector of Bangladesh: "I am sure you have seen that many new products have already come to the market from fishery waste. However, there are scant products in poultry made from fishery waste. Therefore, it is necessary to research how fishery waste can be a protein source for Bangladesh's dairy and poultry industries."



A district fishery officer from Cox's Bazar expressed his knowledge about the potential use of fishery waste: "You know different types of jewelry are made from fish scales. The bones and flesh of fishes are used in developed countries to produce peptides and calcium powder. Chitin and chitosan are being made with the shrimp shell used in makeup products. However, we have not produced those products commercially in Bangladesh, although we have raw materials like fishery waste. Still, we must import it from outside. So, the Bangladesh Government should take steps to utilize these raw materials that are coming as fishery waste and establish new industries to make new products from this fishery waste."

A senior scientific officer from the Bangladesh Fisheries Research Institute (BFRI) in Cox's Bazar expressed his ideas about the future potential of fishery waste: "Researchers and policymakers should work collaboratively to produce more commercial products, like gelatin, leather, bags, ornaments, fish powder, chitin, fish oil, fish sauce, capsules, and surgical gloves or even burnt skin products in Bangladesh. There is a crying need to establish a fishery waste management industry in Bangladesh so that Bangladesh can utilize the huge fishery waste resources and earn foreign currency. It will be helpful economically as well as environmentally. The Bangladesh Government and non-governmental organizations should think about that."

A district fisheries officer from Cox's Bazar expressed his perceptions of the future potential of fishery waste as follows: "There is a huge potentiality of fishery waste in Cox's Bazar. As the market demand rose, approximately six fish-processing industries managed all their fishery waste and valuable parts exported to other countries. The fish bladders of valuable fish species are sold in the international market according to their size and weight. Suppose that the necessary monetary and logistic support can be assured. In that case, there is a possibility to produce margarine, omega-3 fatty acids, gelatin, chitin, chitosan, natural pigments, cosmetics, enzyme isolation, animal feed, poultry feed, fish feed, fertilizer, and biodiesel from the fishery waste of this southern region in Bangladesh."

We also discussed the barriers to or constraints on the sustainable utilization of fishery waste in Bangladesh with the respondents. The key findings of the in-depth interviews follow.

A senior managerial staff member of a seafood-processing industry from Chittagong stated the challenges and way forward in utilizing fishery waste as follows: "Right now the exact market value cannot be predicted as such a kind of industry or market has not developed in Bangladesh yet. However, we should be positive because it has found a wide demand abroad. We would obtain a strong economic figure if it could be used properly. The entrails of the fish that are cut in our house are used as animal food. So, we can collect this waste from fish markets, especially in Dhaka's and Chittagong's major markets. It is possible to produce new products by recycling old ones. Proper utilization of fish waste can create a pollution-free environment. Remember that the environment is also a part of the economy. If we can properly convert it into a product from waste, then you will see that it will have an economic value as well."

A businessman who is directly engaged in the fishery waste business in the Khulna region expressed his opinions about the limitation of the sustainable utilization of fishery resources: "As far as I am aware, the market value of fishery waste is far higher. Because fishery waste would be a good organic product if it could be preserved properly. Furthermore, at this point, everyone is interested in hearing about organic or natural items. Moreover, is it a good sign that people are returning to the natural products that were previously used? So, fishery waste is not just some kind of fish feed. It can also be used as a natural fertilizer, and it is of excellent quality. So, as a result, it has a wide range of applications, many of which are well known. However, I think there is a shortage of public awareness. Suppose that awareness can be increased via social media and mass media about the various uses of fishery waste. In that case, more people will feel interested in the sustainable utilization of the fishery waste."

An academic from the University of Chittagong shared his views about the economic value of fishery waste in the following manner: "In Bangladesh, fish waste is not only

a major environmental problem but also a huge economic loss. For this reason, better fish waste management is needed to overcome these important issues. Therefore, today, the development of sustainable fish waste management plays a key role, which might prevent the generation of waste as much as possible and use the waste generated as a resource for reuse, recycling, and recovery. Thus, the use of fish by-products could contribute to the development of products with high commercial value and consequently to economic growth."

A summary of the results section, stating which responses/actions (according to their relative weight) can help Bangladesh to cope with the problems in the sustainable utilization of fishery waste, is presented in Table 4. The ranking was based on the frequency and priority of their occurrence in the informants' in-depth interviews and the authors' observations. This ranking was not compiled quantitatively.

**Table 4.** Coping with the problems in the sustainable utilization of fishery waste in Bangladesh.

| Rank | Responses/Actions | Problem Addressed |
|:---:|:---:|:---|
| 1 | Further research/initiative of the circular bioeconomy | Failure to take advantage of the fishery waste. We can earn a huge amount of foreign currency by selling this waste. |
| 2 | Extra income | Waste parts of fish have economic value as fishery waste can be used to produce different things. |
| 3 | Fish feed/poultry feed | Local fish feeds/poultry feeds are made unhygienically from fishery waste. Therefore, it needs special consideration to produce environmentally friendly and nutrient-rich fish and poultry feed. |
| 4 | Organic fertilizer | Fishery waste has multiple uses, including a very high-quality organic fertilizer. However, there is a lack of initiatives to make it popular. |
| 5 | New products through recycling | Gelatin, chitin, new ornaments, and fish powder can be made using fishery waste. However, initiatives to set up such industries are still poor in Bangladesh. |
| 6 | Lack of law | The environment department has not taken any step against the dumping of fish waste into the sea due to a lack of laws. If the government initiates any law against this kind of activity degrading the water quality, we can take legal action. |
| 7 | Fish waste-processing center/waste management | The lack of a fish waste-processing center and waste management system is problematic; it is necessary to upgrade the waste management system by carrying out empirical research and building modern fish waste-processing centers near the fish landing sites. |
| 8 | Appreciation | Fishery waste and by-catches have negative and positive effects on biodiversity and the marine habitat. If these are not disposed of properly, the environment will be polluted. This can be a massive opportunity if it can be utilized sustainably. Otherwise, it can exert a harmful impact on the environment. |

## 4. Discussion

### 4.1. Adaptable Application and Sustainability of Fish Waste

More than 70% of the total fish caught is subjected to further processing before being placed on the market, resulting in the production of large amounts (approximately 20–80%)

of fish waste [67,68]. Typical applications of fish leftovers include but are not limited to animal feed, the production of energy through biodiesel and biogas, natural pigments (e.g., astaxanthin or βcarotene), soil fertilizers, food packaging, and enzyme isolation [69,70]. Several studies have analyzed fish waste's possible uses as it represents a rich source of value-added compounds, including enzymes, bioactive peptides, and biopolymers, with many possible uses in several fields [2]. Marine species, including fish products, are famous for their medical uses and are considered to be an exploitable source of efficacious animal-derived medicinal products (ADMPs) [13]. Fish by-products are a nutritionally important source of proteins, fatty acids, and minerals as their composition is like that of fish fillets and other food products used for consumption. Studies on gilthead sea bream fish species have shown that the skin is the most significant protein source, trimmings and bones are rich in calcium, and the head, intestines, and bones are a good source of lipids [71].

Fish oil is found in the flesh, head, frames, fin, tail, skin, and guts in varying quantities. Generally, fish contains 2–30% fat. About 50% of the bodyweight is generated as waste during the fish-processing operation [72], meaning that there is excellent potential for the valorization of this waste, mainly for human consumption or to produce biodiesel. The global fish oil market was valued at USD 1905.77 million in 2019 and is estimated to reach USD 2844.12 million by 2027 [73].

The present study found that significant quantities and varieties of finfish and shellfish waste are being produced in Bangladesh. It was also evident that the increased demand for exports of fishery waste and seafood waste to other Asian and European countries may contribute to reducing the volume of waste. The study also identified opportunities for a significant amount of fishery waste and fishery by-products to be used to produce sustainable aquaculture, livestock feeds, and fish oil in Bangladesh.

### 4.2. Environmental Impacts of Fishery Waste

Fish-processing waste, especially from shore-based facilities, can cause serious harm to the marine environment in the surrounding area. Although fishery waste is a natural pollutant, fish bio-waste can affect the oxygen levels, salinity, temperature, pH levels, and overall abundance of organisms in seawater [74]. An important feature is the environmental impact that fish waste could have on aquatic ecosystems since the release of organic waste might significantly change the community structure and biodiversity of the benthic assemblages [75].

The environmental problems and loss of nutrients associated with seafood process effluents have been pointed out recently. The seafood industries consider huge volumes of by-catches, solid waste, and effluents to be a burden because of their potential to become environmental hazards. The industry dumps enormous amounts of by-catches into the ocean, while reasonable amounts of solid waste are disposed of in landfills or subjected to incineration [75]. Ocean dumping causes reduced oxygen levels at the bottom of the ocean, the burial or smothering of living organisms, and the introduction of disease into the ecosystem of the seafloor [76]. It has been observed that the disposal of waste from lobster processing costs about $7.5 million annually and presents an environmental burden for lobster processors [77].

The present study provides further evidence of various harmful impacts of fishery waste in the adjacent water bodies and land areas. Some of the problems created include water pollution, fouled beaches, insect infestations, and obnoxious odors. All the respondents agreed that fishery waste, including seafood, pollutes the environment. Waste produced by seafood processors ends up in the surrounding environment and pollutes waterways, rivers, canals, the fresh groundwater supply, and indigenous flora and fauna. In addition, seafood waste contributes significantly to industry-generated organic waste. Furthermore, the respondents stated their concern that the present fishery waste and seafood waste management systems in Bangladesh are not adequate, environmentally friendly, or economically beneficial in terms of sustainability. However, they expressed their perceptions and ideas to mitigate the problem sustainably.

*4.3. Recommendations*

Our results reveal that fishery waste, including the seafood-processing waste from industries, can expand the potential to provide jobs or alternative income-generating tasks for local communities, especially fishers and other workers, during fishing ban seasons. Furthermore, based on the literature review, observation, and interviews with relevant stakeholders, this research has identified the following possible steps or recommendations for the sustainable utilization of fishery waste, including seafood waste, in Bangladesh.

If Bangladesh can preserve and even export a third of the fish residues that it generates, it could earn BDT 80 million in a year by exporting fish remains [61]. In Bangladesh, fish-processing waste as a raw material could be a new item for exporting and a source of foreign currency earnings because different products, from medicine to cosmetics, can be made [68]. However, Bangladesh will not be able to achieve that immediately but should focus on products that are easy to make, such as fishmeal. Furthermore, skills need to be learned from other countries to utilize fishery waste to make different products. Therefore, the government should train people to make different valuable products with fishery waste. In this case, the Bangladesh Marine Fisheries Academy (BMFA), in Chittagong, can play a vital role in building up a skilled workforce in this sector as it has the necessary logistic capacities, including technological facilities.

Bangladesh needs to create new opportunities by involving global entrepreneurs and businesses and learning from the experiences of other countries (Iceland, India, and Thailand). There is an urgent need to raise awareness about the sustainable utilization of fishery waste among the general people, including the stakeholders involved in the fish-processing industries. Mass media and social media can help in this regard. Various documentaries can be shown about the different valuable products that can be made using fishery waste or seafood-processing waste, and workshops can be arranged for people involved in the fish-processing value chain.

Local entrepreneurs and infrastructure developers may explore investing avenues to build fish-processing industries near coastal areas. In addition, financial institutions, including the government and non-governmental banks, should relax their existing money-lending policies to encourage borrowers to take out bank loans, enabling them to set up new industries to produce products that will use fishery waste as a raw material.

NGOs should take the lead in conducting empowerment activities for introducing fishery-waste utilization at the local level and can help in providing new employment opportunities. In addition, local influence groups, such as regional political parties and civil societies, may introduce advocacy activities at the policy level to expand the sort of fishery-waste valorization intended to benefit the poor, working to reduce poverty and conserve natural resources [78].

A comprehensive study is required to characterize the quality of available fish waste, including seafood-processing waste, and address the feasibility of its diversified commercial production in Bangladesh. Initiatives should be taken to collect the maximum possible amount of fish waste produced in the country following different strategies. In this regard, a campaign for public awareness could be an effective strategy. A direct marketing link between the local fish cutters in the local markets and the national dealers can also be established to ensure a higher margin for primary stakeholders. In addition, fishery waste-processing factories can be set up to increase the local uses. The government and associated agencies should also promote this business [65].

From the present study, it was evident that fish waste is destined for the production of fishmeal, fertilizers, and fish oil with low profitability or is utilized as raw material for direct feeding in aquaculture and partly thrown away. Hence, better fish-waste management is needed to overcome environmental issues and simultaneously ensure the full use of biomass for purposes of high commercial value. Governments should proactively build policies and regulations, supporting innovative solutions with adequate infrastructure and services.

Fishery waste and by-products may lead to significant problems in terms of management and environmental impacts. In many countries, particular emphasis is given to exploring the possibility of using by-products of fishing, aquaculture, and traditional fishing rather than disposing of them. Aquafeed production from the waste from fishing and the fish-processing industry could be an essential tool for establishing plants of fishery by-products and discards at the local or regional level [8].

Improving waste utilization is essential for a sustainable industry to prevent or minimize the environmental impact [79]. Waste minimization is one of the best approaches to ensure the sustainability of waste management in the fish-processing production process. The waste minimization approach falls into the principle of cleaner production (PCP). According to the United Nations Environment Programme, PCP is defined as a sustainable integrated environmental impact prevention strategy for processes, products, and services to improve overall efficiency and reduce risks to people and the environment. Applying this concept aims to increase the efficiency of natural resource use, prevent environmental pollution, and reduce the formation of waste at the source [80].

A circular, bio-based economy could provide the pathway to a sustainable future. The use of alternative resources that can replace fossil fuels and renewable processes based on sustainability is essential for future generations. Shifting toward a circular bioeconomy requires deep transformations [81]. In addition, the bio-waste valorization approach plays a fundamental role in bringing circularity to the bioeconomy [82]. In this area, tremendous efforts are underway in the scientific community, with the government's support, and are directed mainly at recovering resources from biological waste [83]. Similar opinions were observed in the present study while interviewing the different stakeholders in fishery-waste utilization, including the academics working in fishery and environmental science.

Indeed, introducing circular bioeconomy concepts into the sustainable utilization of fishery waste in Bangladesh is a valid approach. However, empirical research needs to be carried out to integrate circular bioeconomy concepts to valorize fishery waste economically, socially, and environmentally.

### 4.4. Limitations and Scope for Further Study

Fish and seafood are the second-largest export items in Bangladesh after readymade garments. Every year, a considerable quantity of fish-processing by-products is produced in the fish-processing industries, including the seafood-processing industries. However, information on the amount of fish waste generated in Bangladesh was not readily available. To the best of our knowledge, this is the first qualitative baseline study about fishery waste and fishery by-catches in Bangladesh, the environmental impacts of fishery waste, and the potential uses of, and challenges posed by fishery waste for sustainability. Such a situation makes it difficult to compare the findings of this present study. It is challenging to determine the extent to which the issues related to the sustainable utilization of fishery waste are shared from a broader perspective because of the relatively poor data availability on fishery waste and seafood waste produced by the fish-processing industries and seafood-processing industries in Bangladesh.

Further research would thus be needed to subject the findings of the present study to a more robust quantitative methodology. Furthermore, the essential requirement for circular waste management is knowledge about the material flow [84]. It is necessary to know how much material must be managed to develop sound material cycles [85]. In this regard, further research is vital to collect those data empirically and visualize them via STAN (short for subSTance flow ANalysis), including data from the existing literature and published statistics on fishery waste in Bangladesh.

### 5. Conclusions

The efficient minimization and exploitation of fish waste and its transformation into high-value products are an attractive solution from both an environmental and an economic point of view. Fishery-waste utilization, considered a circular bioeconomy concept, has

great potential in Bangladesh. Since fishery waste has versatile uses in different countries, the applications can also be expanded in Bangladesh. Therefore, besides earning foreign exchange, this industry can play a significant role in creating employment, improving livelihoods, increasing environmental sustainability, and contributing to the national economy.

The current study was based on literature reviews and in-country interviews. The status of fishery waste, its environmental impacts, the potential for its utilization, the challenges, and recommendations for implementing the valorization of fishery waste, including seafood-processing waste, through a circular bioeconomy lens were described. There are knowledge gaps that may need to be filled before fishery-waste valorization can be implemented effectively and managed as a concept of the circular bioeconomy. However, the existing theory, synthesis of respondents' perspectives, and recommendation for the versatile, sustainable utilization of fishery waste presented in this study may provide a starting point for thinking about and generating new ideas and opportunities linked to fishery waste valorization in Bangladesh and other developing countries. Finally, fishery waste, including seafood-processing waste, should be considered no less valuable than the fish itself but a precious and profitable resource capable of bringing health, social, economic, and environmental benefits.

**Supplementary Materials:** The following supporting information can be downloaded at: https://www.mdpi.com/article/10.3390/fishes7020084/s1.

**Author Contributions:** Conceptualization, M.M.H.M.; Data curation, M.M.H.M., M.M.U., M.H.I.R., M.G.A.T., T.H.T. and S.N.; Formal analysis, M.H.I.R., M.G.A.T., T.H.T. and S.N.; Funding acquisition, P.S.; Investigation, M.M.U.; Methodology, M.M.H.M., M.H.I.R., T.H.T. and S.N.; Project administration, M.M.U. and P.S.; Resources, M.M.U.; Supervision, P.S.; Visualization, M.M.H.M., M.H.I.R. and M.G.A.T.; Writing—original draft, M.M.H.M., M.H.I.R., M.G.A.T., T.H.T. and S.N.; Writing—review & editing, M.M.H.M., M.M.U. and P.S. All authors have read and agreed to the published version of the manuscript.

**Funding:** Open access funding provided by University of Helsinki.

**Institutional Review Board Statement:** Not applicable.

**Informed Consent Statement:** Written informed consent has been obtained from the participants to publish this paper.

**Data Availability Statement:** Not applicable.

**Conflicts of Interest:** The authors declare no conflict of interest.

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
