# Peer review of "Sustainable Utilization of Fishery Waste in Bangladesh—A Qualitative Study for a Circular Bioeconomy Initiative"

_fishes, doi:10.3390/fishes7020084_

Round 1
Reviewer 1 Report
Lines 42, 68, 162, 221 etc uses "tons" - you should use SI units of "tonnes" as in line 58, line 220 is "metric tonnes" which is the same as tonnes
There is a lengthy diatribe about circular bioeconomy in the introduction to the manuscript consisting of two long paragraphs - this could be much more concise. Circular economy concept is well known and does not warrant such detailed and repetitive description. What is important is how this concept is relevant to this specific piece of research. Unfortunately, this is not clear at the end of these two paragraphs.
In general the introduction is still too long and wordy and includes irrelevant details to the research, such as the total numbers of aquatic species.
"haor, beel, gher" - are these well known terms?
Line 206 repeats content of Line 61
Overall the introduction could be cut in length by half by reducing the repetition, removing irrelevant details, and using more concise language to ensure the introduction, introduces the study specifically.
The introduction does not introduce the research method being used to answer the research questions and why this method was chosen - what are its advantages and disadvantages?
I like that the introduction lands on three clear research questions, unfortunately the subsequent results that are presented do not really address the three questions - e.g measuring the environmental impacts of the waste. I would suggest the research questions perhaps need a little refining so they better match the results that are delivered from a qualitative survey of personnel involved with an industry.
The method section is much improved with much better detail provided on the sampling methodology - thank you.
The recommendations presented in Table 3 are unreferenced and should be.
The section 4.1 in the discussion I am not sure whether it is referring to the study results for Bangladesh or is just general commentary drawn from the wider literature for the world. The latter seems rather pointless when the study pertains to the situation in Bangladesh as drawn from the results from this study which are up for discussion in this section.
Section 4.2 repeats this pattern - appearing to provide a global summary and not actually discussing the results of the current study until the final few sentences
Section 4.3 is a repeat of the general material in the introduction about circular bioeconomies generally, and very little content relating to a discussion of the actual results of this current study.
Section 4.4 has some great recommendation in it. It would be good to see these recommendations being more strongly connected to the actual results coming from the study. Currently it reads more as an opinion piece rather than presenting recommendations emerging from a well grounded and rigorous study.
The conclusion is unfocused and is like an extension of the unruly discussion. Overall, the discussion and conclusion sections are far too long and are not well connected to the actual results of the research work undertaken and presented in the results section. In my view the manuscript could be halved in size and still present the same research results, potentially with more impact than at present.
There are many English grammatical and word use errors that need to be corrected.
There is no report of human ethics approval for the survey procedure - it would be nice to add that detail.
Author Response
Answer to Reviewer 1
|
Manuscript ID |
fishes-1630756 |
|
Title |
Sustainable utilization of fishery waste in Bangladesh—A qualitative study for a circular bioeconomy initiative |
We would like to thank you for your constructive comments in review of the manuscript. Your comments provided valuable insights to refine its contents. In the revised manuscript, we tried to address the issues raised as best as possible. Below we provide our responses to your query-
Q.1.Lines 42, 68, 162, 221 etc uses "tonnes" - you should use SI units of "tonnes" as in line 58, line 220 is "metric tonnes" which is the same as tonnes
Ans- We thank the Reviewer for the valid issues. We revised the manuscript as per Reviewer advise and SI units tonnes are used in the manuscript.
Q.2.There is a lengthy diatribe about circular bioeconomy in the introduction to the manuscript consisting of two long paragraphs - this could be much more concise. Circular economy concept is well known and does not warrant such detailed and repetitive description. What is important is how this concept is relevant to this specific piece of research. Unfortunately, this is not clear at the end of these two paragraphs.
Ans- In the revised manuscript, we tried to concise the circular economy concept. Further, we added how this concept is relevant to the present study. The section can be read is as follows-
Q.3.In general the introduction is still too long and wordy and includes irrelevant details to the research, such as the total numbers of aquatic species.
Ans- We incorporated Reviewer advise and deleted unnecessary part from the introduction.
Q.4."haor, beel, gher" - are these well known terms?
Ans-To minimize the confusion, we deleted the terms from the manuscript.
Q.5.Line 206 repeats content of Line 61
Ans- We revised the manuscript by deleting repeating part mostly in line 61.
Q.6.Overall the introduction could be cut in length by half by reducing the repetition, removing irrelevant details, and using more concise language to ensure the introduction, introduces the study specifically.
Ans-We revised the manuscript as per Reviewer suggestion and concise the introduction part as advised.
Q.6.The introduction does not introduce the research method being used to answer the research questions and why this method was chosen - what are its advantages and disadvantages?
Ans- We agree with this comment and have now incorporated these suggestions into the revised manuscript. We stated the used method in the introduction and explained details in the materials and methods section of the present manuscript, why we used qualitative methods for this study particularly.
Q.7.I like that the introduction lands on three clear research questions, unfortunately the subsequent results that are presented do not really address the three questions - e.g measuring the environmental impacts of the waste. I would suggest the research questions perhaps need a little refining, so they better match the results that are delivered from a qualitative survey of personnel involved with an industry.
Ans-We revised the manuscript as per Reviewer advise and the research question of the present study is as such-
- What is the present status of fishery waste and fishery by-catches and of their environmental impacts in Bangladesh?
- What are the potential uses and challenges of fishery waste for sustainability?
Q.8.The method section is much improved with much better detail provided on the sampling methodology - thank you.
Ans- We are glad that you accepted the method section as readable.
Q.9.The recommendations presented in Table 3 are unreferenced and should be.
Ans- Based on primary and secondary data, we summarized fishery waste and fishery by-products' general uses and present uses in Bangladesh, followed by recommendations for sustainable utilization of such resources. As the recommendations were based on participants perceptions and authors realization, we did not use reference in this case.
Q.10.The section 4.1 in the discussion I am not sure whether it is referring to the study results for Bangladesh or is just general commentary drawn from the wider literature for the world. The latter seems rather pointless when the study pertains to the situation in Bangladesh as drawn from the results from this study which are up for discussion in this section.
Ans-We tried to highlight the general use of fishery waste worldwide and general commentary are drawn from the wider literature. Later we discussed what is lack of in Bangladesh and what can be done further as a part of discussion. Finally, we revised the manuscript as per Reviewer advise and deleted unnecessary part from the discussion.
Q.11.Section 4.2 repeats this pattern - appearing to provide a global summary and not actually discussing the results of the current study until the final few sentences.
Ans- We tried to highlight the general use of fishery waste worldwide and general commentary are drawn from the wider literature. Later we discussed what is lack of in Bangladesh and what can be done further as a part of discussion. Finally, we revised the manuscript as per Reviewer advise and deleted unnecessary part from the discussion.
Q.12.Section 4.3 is a repeat of the general material in the introduction about circular bioeconomies generally, and very little content relating to a discussion of the actual results of this current study.
Ans- We tried to highlight the general use of fishery waste worldwide and general commentary are drawn from the wider literature. Later we discussed what is lack of in Bangladesh and what can be done further as a part of discussion. Finally, we revised the manuscript as per Reviewer advise and deleted unnecessary part from the discussion.
Q.13.Section 4.4 has some great recommendation in it. It would be good to see these recommendations being more strongly connected to the actual results coming from the study. Currently it reads more as an opinion piece rather than presenting recommendations emerging from a well grounded and rigorous study.
Ans- We revised the manuscript as per Reviewer advise and tried our best to connect the recommendations with the results of the present study.
Q.14.The conclusion is unfocused and is like an extension of the unruly discussion. Overall, the discussion and conclusion sections are far too long and are not well connected to the actual results of the research work undertaken and presented in the results section. In my view the manuscript could be halved in size and still present the same research results, potentially with more impact than at present.
Ans-We revised the conclusion section as per Reviewer advise and concise more.
Q.15.There are many English grammatical and word use errors that need to be corrected.
Ans- We revised the manuscript with the help of a professional proof-reader and a certificate is attached.
Q.1.There is no report of human ethics approval for the survey procedure - it would be nice to add that detail.
Ans- We revised the manuscript by adding few sentences of human ethics approval for the survey procedure in the method sections. Also, a copy of a consent form is attached as a supplementary document.
Answer to Reviewer 1
|
Manuscript ID |
fishes-1630756 |
|
Title |
Sustainable utilization of fishery waste in Bangladesh—A qualitative study for a circular bioeconomy initiative |
We would like to thank you for your constructive comments in review of the manuscript. Your comments provided valuable insights to refine its contents. In the revised manuscript, we tried to address the issues raised as best as possible. Below we provide our responses to your query-
Q.1.Lines 42, 68, 162, 221 etc uses "tonnes" - you should use SI units of "tonnes" as in line 58, line 220 is "metric tonnes" which is the same as tonnes
Ans- We thank the Reviewer for the valid issues. We revised the manuscript as per Reviewer advise and SI units tonnes are used in the manuscript.
Q.2.There is a lengthy diatribe about circular bioeconomy in the introduction to the manuscript consisting of two long paragraphs - this could be much more concise. Circular economy concept is well known and does not warrant such detailed and repetitive description. What is important is how this concept is relevant to this specific piece of research. Unfortunately, this is not clear at the end of these two paragraphs.
Ans- In the revised manuscript, we tried to concise the circular economy concept. Further, we added how this concept is relevant to the present study. The section can be read is as follows-
Q.3.In general the introduction is still too long and wordy and includes irrelevant details to the research, such as the total numbers of aquatic species.
Ans- We incorporated Reviewer advise and deleted unnecessary part from the introduction.
Q.4."haor, beel, gher" - are these well known terms?
Ans-To minimize the confusion, we deleted the terms from the manuscript.
Q.5.Line 206 repeats content of Line 61
Ans- We revised the manuscript by deleting repeating part mostly in line 61.
Q.6.Overall the introduction could be cut in length by half by reducing the repetition, removing irrelevant details, and using more concise language to ensure the introduction, introduces the study specifically.
Ans-We revised the manuscript as per Reviewer suggestion and concise the introduction part as advised.
Q.6.The introduction does not introduce the research method being used to answer the research questions and why this method was chosen - what are its advantages and disadvantages?
Ans- We agree with this comment and have now incorporated these suggestions into the revised manuscript. We stated the used method in the introduction and explained details in the materials and methods section of the present manuscript, why we used qualitative methods for this study particularly.
Q.7.I like that the introduction lands on three clear research questions, unfortunately the subsequent results that are presented do not really address the three questions - e.g measuring the environmental impacts of the waste. I would suggest the research questions perhaps need a little refining, so they better match the results that are delivered from a qualitative survey of personnel involved with an industry.
Ans-We revised the manuscript as per Reviewer advise and the research question of the present study is as such-
- What is the present status of fishery waste and fishery by-catches and of their environmental impacts in Bangladesh?
- What are the potential uses and challenges of fishery waste for sustainability?
Q.8.The method section is much improved with much better detail provided on the sampling methodology - thank you.
Ans- We are glad that you accepted the method section as readable.
Q.9.The recommendations presented in Table 3 are unreferenced and should be.
Ans- Based on primary and secondary data, we summarized fishery waste and fishery by-products' general uses and present uses in Bangladesh, followed by recommendations for sustainable utilization of such resources. As the recommendations were based on participants perceptions and authors realization, we did not use reference in this case.
Q.10.The section 4.1 in the discussion I am not sure whether it is referring to the study results for Bangladesh or is just general commentary drawn from the wider literature for the world. The latter seems rather pointless when the study pertains to the situation in Bangladesh as drawn from the results from this study which are up for discussion in this section.
Ans-We tried to highlight the general use of fishery waste worldwide and general commentary are drawn from the wider literature. Later we discussed what is lack of in Bangladesh and what can be done further as a part of discussion. Finally, we revised the manuscript as per Reviewer advise and deleted unnecessary part from the discussion.
Q.11.Section 4.2 repeats this pattern - appearing to provide a global summary and not actually discussing the results of the current study until the final few sentences.
Ans- We tried to highlight the general use of fishery waste worldwide and general commentary are drawn from the wider literature. Later we discussed what is lack of in Bangladesh and what can be done further as a part of discussion. Finally, we revised the manuscript as per Reviewer advise and deleted unnecessary part from the discussion.
Q.12.Section 4.3 is a repeat of the general material in the introduction about circular bioeconomies generally, and very little content relating to a discussion of the actual results of this current study.
Ans- We tried to highlight the general use of fishery waste worldwide and general commentary is drawn from the wider literature. Later we discussed what is lack of in Bangladesh and what can be done further as a part of the discussion. Finally, we revised the manuscript as per the Reviewer's advice and deleted unnecessary parts from the discussion.
Q.13.Section 4.4 has some great recommendation in it. It would be good to see these recommendations being more strongly connected to the actual results coming from the study. Currently it reads more as an opinion piece rather than presenting recommendations emerging from a well grounded and rigorous study.
Ans- We revised the manuscript as per Reviewer's advice and tried our best to connect the recommendations with the results of the present study.
Q.14.The conclusion is unfocused and is like an extension of the unruly discussion. Overall, the discussion and conclusion sections are far too long and are not well connected to the actual results of the research work undertaken and presented in the results section. In my view the manuscript could be halved in size and still present the same research results, potentially with more impact than at present.
Ans-We revised the conclusion section as per the Reviewer's advice and concise.
Q.15.There are many English grammatical and word use errors that need to be corrected.
Ans- We revised the manuscript with the help of a professional proofreader and a certificate is attached.
Q.16.There is no report of human ethics approval for the survey procedure - it would be nice to add that detail.
Ans- We revised the manuscript by adding a few sentences of human ethics approval for the survey procedure in the method sections. Also, a copy of a consent form is attached as a supplementary document.

Reviewer 2 Report
The subject is very important to the fish sector. I believe that the papper is the first step to bring informations about the fish waste generation and utilization.
I recomend the formatation reviwe in the table 3.
I recomend to present the data from the interviews in the pag 11,12,13,14,15... in another more formal and scientific format
Author Response
Answer to Reviewer 2
|
Manuscript ID |
fishes-1630756 |
|
Title |
Sustainable utilization of fishery waste in Bangladesh—A qualitative study for a circular bioeconomy initiative |
We would like to thank you for your constructive comments in review of the manuscript. Your comments provided valuable insights to refine its contents. In the revised manuscript, we tried to address the issues raised as best as possible. Below we provide our responses to your query-
Q.1. I recomend the formatation reviwe in the table 3.
Ans- We revised the manuscript and changed the format of Table 3 as per Reviewer's advice.
Q.2. I recomend to present the data from the interviews in the pag 11,12,13,14,15... in another more formal and scientific format.
Ans- We thank the Reviewers for this valid point. However, we have presented the results in a format by following the criteria and previously published articles in the MDPI journal that used the qualitative study as a method.

This manuscript is a resubmission of an earlier submission. The following is a list of the peer review reports and author responses from that submission.
Round 1
Reviewer 1 Report
Dear Authors,
I found this manuscript interesting and with a lot of information that give it the aspects of a review article rather than a research article. The writing needs an English language revision made by a native speaker. Moreover, the organization of the manuscript needs to be revised entirely, trying to give it the aspects of a research article, addressing some points as suggested:
The entire manuscript is too wordy, try to be more concise in almost all its parts, or write a review article.
2. Theoretical approaches- Circular Bioeconomy: this paragraph shouldn't exist in a normal research article. Move this information within Introduction, trying to be more concise.
Material and Methods section needs to be enriched by practical information on the research part of this study. The administered questionary was missed, please add it. The information on the software used for data elaboration were missed, please add it.
Results section needs, at least, a resumable table with the key results of the interviews, such as of the anonymous interviewed (age class, year of experience, fishery wastes daily relations, ecc..), in the present form this manuscript have no results for the reader. The use of proper statistical index to give more resonance to your results, it's highly suggested.
Discussion are really too wordy, try to be more concise and effective through the key results of your survey.
Other minor points to address were:
Some Keywords are already present in Title, this should be avoided to give more resonance to your manuscript during the web search phases. Try to substitute it with others.
References are not always well referenced, please double check it.
Best regards
The Reviewer
Reviewer 2 Report
The overall quality of English language in this manuscript is poor. For example there are sentences for which it is hard to decipher the intended meaning. The introduction is a long, rambling and quite repetitive section of the manuscript that does not do a good job of introducing the research that follows. The implication inherent in the introductory content is that Bangladeshi fisheries are poor at waste management, but the global data presented of waste in fisheries suggests that this is a world-wide problem and not just one faced by Bangladesh. The focus of the introduction is lacking, with quite a lot of description of circular bioeconomy concepts, but it is not clear how this actually relates to the research that is about to be presented. It would be useful for international audiences to characterise Bangladeshi fisheries, quantities of fish landed, diversity of species landed, type of fishing used - is it industrial trawling or artisinal fishing from coastal villages.
The introduction poses three research aims - none of which are adequately addressed in the subsequent research and none of them can be answered given the constraints of the research approach that has been used for this study, i.e.,
"What is the present status of fishery waste and fishery by-catch in Bangladesh?
What are the environmental impacts of fishery waste?
What is the potential uses and challenges of fishery waste for sustainability?"
The research methods appear limited in scope - given the scale and diversity of Bangladeshi fisheries interviewing just two fishers from each of three regions is insufficient to gain an understanding of a major source of waste - discarded bycatch - as this small sample of fishers are unlikely to represent the wider extent of fish species and fishing methods that occur in the country. Bycatch can only really be reported by fishers on the water who are familiar with the practice and not by fisheries managers and fish processors who rarely, if ever, observe discarding in practice.
The "qualitative research" consists of a series of researcher selected quotes the basis for selection is unclear and there is no systematic analyses of the survey information provided other than the breakdown of interviewee sources. The research does not appear to follow fundamental approaches used in qualitative research practice or fails to explain how these approaches are being applied in the case of this study.
The discussion and conclusions appear to emerge from the preconceived theoretical framework advanced in the introduction and are not directly supported by the results of the research. The research aims, as stated in the introduction, are not resolved by the research methods, results or conclusions.
In my view the research methodology has insufficient survey sample size and coverage to adequately address the research aims as proposed by the authors. The survey does provide some insights into the situation and views of fishing industry personnel in relation to the issue of waste. This has value in itself, and if presented in a more rigorous research framework would provide a valuable research contribution, as many nations face the challenge of understanding and improving the management of waste from their fisheries. However, to achieve this the authors' would need to significantly revise the manuscript to improve its focus and quality of presentation.